# Chemical Variability of Peel and Leaf Essential Oils in the *Citrus* Subgenus *Papeda* (Swingle) and Few Relatives

**DOI:** 10.3390/plants10061117

**Published:** 2021-05-31

**Authors:** Clémentine Baccati, Marc Gibernau, Mathieu Paoli, Patrick Ollitrault, Félix Tomi, François Luro

**Affiliations:** 1Laboratoire Sciences Pour l’Environnement, Equipe Chimie et Biomasse, Université de Corse—CNRS, UMR 6134 SPE, Route des Sanguinaires, 20000 Ajaccio, France; clementine.baccati@gmail.com (C.B.); gibernau_m@univ-corse.fr (M.G.); paoli_m@univ-corse.fr (M.P.); 2UMR AGAP Institut, Université Montpellier, CIRAD, INRAE, Institut Agro, 20230 San Giuliano, France; patrick.ollitrault@cirad.fr (P.O.); francois.luro@inrae.fr (F.L.); 3CIRAD, UMR AGAP, 20230 San Giuliano, France

**Keywords:** *C. latipes*, *C. macroptera*, *C. ichangensis*, *C. micrantha*, *C. hystrix*, diversity

## Abstract

The *Papeda Citrus* subgenus includes several species belonging to two genetically distinct groups, containing mostly little-exploited wild forms of citrus. However, little is known about the potentially large and novel aromatic diversity contained in these wild citruses. In this study, we characterized and compared the essential oils obtained from peels and leaves from representatives of both *Papeda* groups, and three related hybrids. Using a combination of GC, GC-MS, and ^13^C-NMR spectrometry, we identified a total of 60 compounds in peel oils (PO), and 76 compounds in leaf oils (LO). Limonene was the major component in almost all citrus PO, except for *C. micrantha* and *C. hystrix*, where β-pinene dominated (around 35%). LO composition was more variable, with different major compounds among almost all samples, except for two citrus pairs: *C. micrantha*/*C. hystrix* and two accessions of *C. ichangensis*. In hybrid relatives, the profiles were largely consistent with their *Citrus*/*Papeda* parental lineage. This high chemical diversity, not only among the sections of the subgenus *Papeda*, but also between species and even at the intraspecific level, suggests that *Papeda* may be an important source of aroma diversity for future experimental crosses with field crop species.

## 1. Introduction

*Citrus* species are native to Southeast Asia, and their exceptional diversity is the result of both migration and geographical isolation over the course of their evolutionary history [1]. The current cultivated forms are the result of crosses between species that evolved in Southeast Asia [2,3,4,5]. Two other genera capable of crossing with *Citrus* species are also found in Asia: *Fortunella* and *Poncirus*. There are also Oceanian species known to hybridize with *Citrus* species, belonging to the genera *Eremocitrus*, *Microcitrus*, *Clymenia*, and *Oxanthera*. Together, these genera constitute the “true citrus” group as described by Swingle and Reece [6]. One of the main characteristics of *Citrus* is the presence of highly aromatic essential oils in tissue storage cells of the fruit, leaf, and flower (petals). These essential oils are complex mixtures that can contain hundreds of compounds with a very wide chemical diversity, which is prized by the aromatic and cosmetic industry [7]. The composition of essential oils in the majority of citrus fruits grown for consumption is very well documented [8]. However, much of the aromatic diversity found in other *Citrus*, including fruits of the *Papeda* subgenus, remains largely unknown.

Swingle recognized two subgenera in *Citrus*: *Papeda* and *Citrus* [9]. In the subgenus *Papeda*, he defined two sections (*Papeda* and *Papedocitrus*) with four species (*C. hystrix* D.C., *C. macroptera* Montrouz, *C. micrantha* Wester, and *C. celebica* Koord) listed in the former, and two species (*C. ichangensis* Swingle (Ichang papeda) and *C. latipes* (Swingle) Tanaka (Khasi papeda)) in the latter [9]. The section *Papedocitrus* is considered as intermediate between the two subgenera *Papeda* and *Citrus*. In the chapter entitled “Botany of citrus and its wild relatives”, Swingle and Reece described the subgenus *Papeda* as follows: “*pulp-vesicles containing numerous droplets of acrid oil; petioles long and very broadly winged, but not cordate, often nearly as broad as the leaf blades; stamen usually free [..] flowers larger and petioles very long, 1.75–3 longer than broad*” [6]. Recently, exploration of the *Citrus* genome by molecular markers and sequencing has demonstrated that *Papeda* is a non-homogeneous group actually consisting of two very distinct (polyphyletic) genetic groups. The first group includes *C. micrantha* as reference, and the second is represented by *C. cavaleriei* H. Lév. Ex Cavalerie (or *C. ichangensis*) [1,10]. These two genetic groups are considered to be two ancestral species, that have generated some cultivated varieties such as Yuzu (*C. ichangensis* × *C. reticulata*), Alemow and Mexican lime (*C. micrantha* × *C. medica*), Nasnaran mandarin (*C. micrantha* × *C. reticulata*), or Ichang lemon (*C. maxima* × *C. junos*) through outcrossing with other ancestral species (*C. maxima* (Burm.) Merr, *C. reticulata* Blanco, *C. medica* L.) [1,7,11,12].

Recently, Ollitrault et al. [13] proposed a new classification taking into account phylogenetic relationships and sexual compatibility, building on the former classifications of Tanaka [14], Swingle and Reece [6], and Zhang and Mabberley [15]. In the Papeda group, two true species are now recognized. The first, *C. cavaleriei*, originates from West-Central and Southwestern China, and includes *C. ichangensis* and *C. latipes*. The second, *C. hystrix*, originates from Southern Philippines and includes *C. micrantha* (with two varieties, micrantha (Biasong) and microcarpa (Samuyao), and appears to be very closely related to *C. micrantha* [15]. *C. hystrix* (Combava) also appears very closely related to *C. micrantha* [15]. This new classification partially confirms the work of Swingle and Reece, who had divided the Papeda group into two sections. However, it should be noted that the classification of Melanesian papeda (*C. macroptera*) has not yet been considered in this phylogenomic taxonomy. In terms of genetic diversity, there are very few studies concerning the *Papeda* group, though high intraspecific genetic diversity has been identified in *C. macroptera* [16] and *C. cavaleriei* (or *C. ichangensis*) [17].

Data concerning the chemical composition of peel and leaf oils from *Citrus* classified as *Papeda* are scarce; the literature is mainly focused on cultivated hybrids such as Yuzu [18,19]. Leaf oil composition was also reported in *C. ichangensis* [20,21,22]. *C. hystrix* is also well described in the literature [21,22,23,24], while *C. macroptera* leaf oil was described by Huang et al. [19] and Waikedre et al. [24]. *C. latipes* and *C. macrophylla* leaf oils were described only once, in the same publication [25]. The lack of data on *C. macrophylla* may be due to its sole use as a rootstock for citrus cultivation [26]. To our knowledge, there is no chemical data concerning *C. micrantha* in the literature.

The chemical composition of peel and leaf essential oils can be determined by (i) gas chromatography (GC) retention indices (RI) calculated for polar and apolar columns and (ii) gas chromatography-mass spectrometry (GC-MS). These methods provide abundant information, not only for metabolism-related research, but also for chemotaxonomy. Consequently, several studies on *Citrus* have used this approach [27,28]. The chemotaxonomy of Mangshanyegan (*C. nobilis* Lour.), was determined by comparison of volatile profiles of fruits and leaves and those of 29 other genotypes of *Citrus*, *Poncirus*, and *Fortunella* [20]. The chemical components identified in the peels of 66 citrus germplasms from four *Citrus* horticultural groups (mandarin, orange, grapefruit, and lemon) were also used for biomarker mining. Thirty potential biomarkers were identified, and four compounds (β-elemene, valencene, nootkatone, and limettin) were validated as biomarkers [29]. However, Luro et al. [30] found that the diversity based on leaf oil compositions from *Citrus medica* varieties did not agree with the molecular diversity and was therefore unsuitable for intraspecific phylogenetic studies.

In this context, the aim of this study was to investigate the diversity of chemical composition of peel and leaf oils from *Citrus* belonging to the subgenus *Papeda* present in the INRAE-CIRAD citrus germplasm bank (Corsica, France). All accessions are fully indexed in a plot with identical climatic and agronomic growing conditions [31]. These conditions are suitable to study the relationship between chemistry and taxonomy and to produce reference data for Papeda peel and leaf oils composition. We analyzed ten samples from eight species, including three *Papeda* species (*C. hystrix*, *C. micrantha*, and *C. macroptera*) and two *Papedocitrus* species (three accessions of *C. ichangensis* and one of *C. latipes*) to characterize the two sections of the subgenus *Papeda*, and three related hybrid species (*C. junos*, *C. macrophylla*, and *C. wilsonii*) in order to investigate the inheritance of chemical characters. To evaluate chemical diversity within and among species, we performed Principal Component Analysis on the peel and leaf oil data.

## 2. Results and Discussion

### 2.1. Peel Oils

Among the ten studied accessions, only nine peel oil samples were obtained by hydrodistillation of peels, because one accession of *C. ichangensis* did not produce a sufficient number of fruits. In total, 60 compounds were identified in peel oils, accounting for 91.6% to 99.9% of the total oil composition (Table 1).

All of the peel oil samples were dominated by monoterpene hydrocarbons, mostly due to the abundance of limonene (20.7–81.4%), as described in the literature [23]. Despite this common characteristic, several chemical profiles were observed.

#### 2.1.1. Section *Papeda*

*C. hystrix* and *C. micrantha* showed low contents of limonene (respectively, 25.2 and 20.7%), and are associated with higher β-pinene contents, which was the major component (35.0 and 33.4%). While *C. hystrix* oil showed a large amount of sabinene (22.7%), *C. micrantha* oil contained a noticeable quantity of oxygenated monoterpenes with citronellol (6.8%), α-terpineol (6.6%), terpinen-4-ol (3.8%), citronellyl acetate (3.1%), and other smaller components. These two samples were clearly discriminated on PCA analysis (Figure 1). According to the literature, this low content of limonene is typical in *C. hystrix*. A review by Lawrence [32] gave the following main components: β-pinene (20.4–42.2%), sabinene (13.0–25.9%), citronellal (3.4–16.8%), limonene (2.8–14.2%), terpinen-4-ol (3.8–8.9%), and α-terpineol (1.7–7.4%).

*C. macroptera* oil contained limonene (53.8%), sabinene (12.4%), and β-pinene (3.9%), as well as monoterpene alcohols such as linalool (11.8%) and terpinen-4-ol (4.3%). As observed for *C. hystrix* and *C. micrantha*, the percentage of oxygenated monoterpenes was elevated (18.6%). This composition of *C. macroptera* peel oil is different than the ones described by Rana and Blazquez [33], which contained 55.3% limonene, 4.7% of (*E*)-β-caryophyllene, and 3.5% geraniol; and Miah et al. [34], which contained limonene (73.5%), δ-cadinene (3.4%), and α-terpineol (3%). This compositional diversity of *C. macroptera* peel oil is consistent with the genetic diversity revealed by DNA markers [16].

#### 2.1.2. Section *Papedocitrus*

*C. ichangensis* and *C. latipes* oils were characterized by medium percentages of limonene varying between 42.3 and 58.2%. However, a strong chemical variability was observed for the *Papedocitrus* section (Figure 1). The *C. ichangensis* accession *ich-3* exhibited a typical composition, dominated by monoterpene hydrocarbons, limonene (58.2%), sabinene (9.6%), β-phellandrene (8.8%), and *p*-cymene (4.4%), and mostly terpinen-4-ol (7.3%) for the oxygenated compounds. The accession *ich-2* had an atypical composition, characterized by a 1:1 ratio of monoterpenes (42.8% of hydrocarbons and 1.9% of oxygenated monoterpenes) and sesquiterpenes (33.4% of hydrocarbons and 7.3% of oxygenated sesquiterpenes), as well as a noticeable quantity of acyclic compounds (6.0%). Moreover, the percentages of β-bisabolene (18.4%) and intermedeol (4.7%) were notable, as were other sesquiterpenes identified in smaller proportions such as *trans*-α-bergamotene (3.2%), γ-muurolene (3.1%), valencene (2.7%), and (*E*)-nerolidol (1.6%). For comparison, a recent study found that among several *C. ichangensis* peel oils, one of them contained higher amounts of sesquiterpenes than monoterpenes [21]. In this study, α-cadinene, β-bourbonene, and the acyclic esters butyl butanoate and ethyl hexanoate were reported in *C. ichangensis* peel oil. Another study described the composition of *C. ichangensis* peel oil, with higher percentages of limonene (61.0–70.4%) but similar amounts of β-bisabolene (9.3–13.0%) and (*E*)-nerolidol (3.1–3.9%) [20]. Our results are therefore consistent with published results on the presence of both acyclic compounds and a large range of sesquiterpene hydrocarbons in *C. ichangensis*.

*C. latipes* peel oil composition differed from those of *C. ichangensis* by exceptionally high percentages of myrcene (18.8%) and γ-terpinene (16.2%). No chemical data were found about peel oil of *C. latipes* in the literature.

#### 2.1.3. Related Species

Peel oils of *C. wilsonii*, *C. junos*, and *C macrophylla* were characterized by high limonene contents (66.9, 79.9, and 81.4%, respectively), and were associated with noticeable amounts of γ-terpinene (10.1, 8.8, and 5.0%, respectively). The chemical composition of *C. wilsonii* peel oil was close to one previously reported from a hexane extract, which contained limonene (56.6%), γ-terpinene (17.8%), β-phellandrene (3.8%), β-pinene (2.4%), linalool (1.6%), and myrcene (1.3%) [7]. In this study, the authors concluded that *C. wilsonii* combined three ancestral genomes (*C. maxima*, *C. ichangensis*, and *C. reticulata*) and may be a pummelo × Yuzu hybrid. Based on the high percentage of limonene usually found in peel oils, it is quite difficult to evaluate the inheritance of these accessions only based on this characteristic.

The chemical composition we observed for *C. junos* is similar than those described by Dugo and Di Giacomo [23] with the proportion of limonene varying between 60.4 and 82.4%, mainly associated with γ-terpinene (7.6–10.7%) and linalool (0.9–5.6%). No data were found about peel oil composition of *C. macrophylla*.

Essential oil compositions dominated by limonene are frequently observed in many *Citrus* species, such as *C reticulata* [35] (around 70%) and *C. sinensis* (L.) Osbeck, *C. aurantium* L., *C.* × *paradisi* Macfad., and *C. aurantiifolia* (Christm.) Swingle [23] (around 90%). Citron (*C. medica*) peel oils are known to contain variable amounts of limonene (39.5–94.3%), either as the only major component or associated with geranial/neral or γ-terpinene [36]. Similarly, lemon (*C. limon* (L.) Burm.) peel oils can contain uneven quantities of limonene (38.1–95.8%), occasionally in association with other major components including γ-terpinene, linalool, β-pinene [37]. Nevertheless, the proportion of limonene in peel essential oil of citrons (*C. medica*) and lemons (*C. limon*) is lower (between 40 and 50%) [23,30]. Two of the three *Citrus* × *Papeda* hybrids, Alemow and Yuzu, have a higher proportion of limonene in the peel essential oil than their two respective *Citrus* parents. These are two cases of transgressive inheritance, previously observed in a clementine × mandarin population [38].

Yuzu and Ichang lemon, hybrids of *C. ichangensis* and other *Citrus* species, exhibited a significant amount of β-phellandrene. Such proportions at rates higher than 1% are rather unusual in *Citrus*, and could be inherited from a parent with a chemical profile close to the *ich-3* accession of *C. ichangensis,* which expressed a higher amount of this compound (8.8%) than other species.

In our sampling, *Papedocitrus* peel oils constituted an intermediate between low amounts of limonene observed in *Papeda* section (around 25%) and high percentages (around 80%) in the related species. The variability of the chemical profiles is very large within the *Papeda* samples, as seen in Figure 1, a PCA in which the two principal axes accounted for 81.8% (70.9 and 10.7%; F1 and F2, respectively). This diversity is mainly due to three compounds (limonene, β-pinene, and sabinene) that separate the *micrantha*/*hystrix* pair from all other citrus fruits in general, but especially to *C. macrophylla* and *C. junos*, which show characteristics of their parent of the *Citrus* subgenus (citron and mandarin) (Appendix A).

### 2.2. Leaf Oils

In total, 76 compounds were identified in leaf oils, accounting for 93.6% to 99.3% (Table 2). The yields of the ten leaf oil samples varied drastically between 0.015 to 0.18% (Table 2). For example, the three *C. ichangensis* samples exhibited very different yields and strong intraspecific variability in composition.

The ten leaf oil samples exhibited a chemical composition dominated by monoterpenes, as usually found in *Citrus* leaf essential oils [23]. However, we observed substantial quantitative variability among the major components: sabinene (0–44.6%), β-pinene (0–15.7%), (*Z*)-β-ocimene (tr–18.2%), (*E*)-β-ocimene (0.2–62.7%), γ-terpinene (0–28.2%), linalool (0.2–24.6%), citronellal (0–78.1%), neral (0–18.9%), geranial (0–24.7%).

*C. hystrix*, *C. micrantha*, and *C. macrophylla* leaf oils were dominated by oxygenated monoterpenes, whereas *C. junos*, *C. ichangensis* (three accessions), and *C. macroptera* were dominated by monoterpene hydrocarbons. The last species, *C. latipes*, exhibited a nearly 1:1 ratio between hydrocarbon/oxygenated terpenes. 

#### 2.2.1. Section *Papeda*

##### Combava (*C. hystrix* DC.) and Biasong (*C. micrantha* Wester)

*C. hystrix* and *C. micrantha* leaf oils exhibited a close chemical composition strongly dominated by citronellal (respectively, 78.1 and 76.1%) and its derivatives, citronellol (3.4 and 4.4%), and citronellyl acetate (0.7 and 5.1%). These two samples were also highly discriminated in the PCA (Figure 2).

Similar compositions were previously reported for *C. hystrix* oils: citronellal between 58.9 and 81.5%, citronellol between 6.0 and 8.2%, and citronellyl acetate between 0.9 and 5.1% [23]. A recent review on *C. hystrix* found that some authors described leaf oils with 1.4 to 72.5% citronellal, while others described leaf oil dominated by limonene (40.7–83.9%) [39]. A New-Caledonian study showed a drastically different chemical composition of *C. hystrix* leaf oil, dominated by terpinen-4-ol (13.0%), β-pinene (10.9%), α-terpineol (7.6%), and citronellol (6.0%) with a very low content in citronellal (2.7%) [24]. Finally, Zhang et al. [22] showed four accessions of *C. hystrix*, with the three same major components (citronellal, geranial, and geranyl acetate), but in different relative quantities. To our knowledge, there is no existing description of *C. micrantha* oil in the literature.

It is interesting to note that in many phylogenetic studies, *C. micrantha* and *C. hystrix* are grouped together or have even formed a separate cluster [40,41]. These studies seem to indicate that in this case, genetics and chemistry agree in considering *C. micrantha* and *C. hystrix* as related species.

##### Melanesian Papeda (*C. macroptera* Montr.)

The leaf oil of *C. macroptera* is characterized by large amounts of sabinene (32.4%), β-pinene (15.7%), and linalool (18.2%) as well as significant percentages of (*E*)-β-ocimene (8.6%) and terpinen-4-ol (3.8%). Two articles reported the chemical composition of this essential oil. The first reported that sabinene (20.9%) predominated, in association with geranyl acetate (15.5%), β-phellandrene (9.1%), geranial (8.7%), (*E*)-β-ocimene (8.0%), and neral (6.8%) [19]. Conversely, hydrocarbons were the main components in the second study: β-pinene (33.3%), α-pinene (25.3%), *p*-cymene (17.6%), and (*E*)-β-ocimene (6.7%), with very little sabinene (4.8%) and no geranyl acetate [24]. Therefore, the chemical composition we identified in this study is novel, suggesting significant variability in this species, as previously observed with DNA molecular markers [16].

#### 2.2.2. Section Papedocitrus

##### Ichang Papeda (*C. ichangensis* Swingle)

The essential oils of *C. ichangensis* showed significant intraspecific variation (Figure 2). Two accessions, *ich-1* and *ich-2*, were characterized by a dominance of (*Z*)/(*E*)-β- ocimenes, in variable amounts (18.2%/62.7% and 13.0%/32.4%, respectively). The first accession *ich-1* also showed appreciable amounts of linalool (9.3%) and linalyl acetate (10.8%). Moreover, these ocimene-type oils contained an appreciable amount of alismol (1.7 and 1.6%), an unusual sesquiterpene in *Citrus*. Indeed, occurrences of alismol in *Citrus* oils have already been found in kumquats (*Fortunella* genus) [42] and in *C.* × *jambhiri* Lush. [43]. The third accession *ich-3* is completely different, and is dominated by sabinene (44.6%), in addition to β-phellandrene (11.7%) and terpinen-4-ol (8.4%).

In a recent study, Zhang et al. [22] reported the chemical composition of ten accessions of *C. ichangensis*. Five of ten oil samples exhibited the two aforementioned ocimenes as major components, in addition to linalyl acetate in four accessions and α-pinene in another one, whereas two of ten oil samples were dominated by sabinene, in addition to with γ-terpinene and limonene. In our sampling, a similar 2:1 ratio between these two compositions was observed. The three other accessions of *C. ichangensis* in Zhang et al. [22] were dominated by γ-terpinene in two samples, and linalyl acetate for the final sample. Moreover, the authors indicated that percentages of sesquiterpene hydrocarbons such as (*E*)-β-caryophyllene, (*E*)-β-farnesene, β-elemene, or germacrene D were occasionally high, as observed in our sampling.

Another *Citrus* oil known to contain an appreciable amount of ocimene is a lemon named “Poire du Commandeur” or “Peer lemon” (*C.* × *lumia* Risso and Poit.), a purported pummelo × mandarin hybrid [12] characterized by high contents of β-pinene (41.4%) and (*E*)-β-ocimene (15.8%), associated with linalool (11.2%), limonene (8.6%), and sabinene (4.8%) [37].

The strong intraspecific diversity observed at the level of aromatic compounds is in agreement with the high genetic diversity of this taxa observed at the DNA level [17].

##### Khasi papeda (*C. latipes* (Swingle) Tanaka)

This species was characterized by a nearly 1:1 hydrocarbon/oxygenated terpenes ratio. The oil sample was dominated by limonene (41.0%), associated with linalool (24.6%) and citronellal (14.1%). We observed that citronellal (14.1%), citronellol (1.8%), and citronellyl acetate (1.0%), the main components of *C. hystrix* and *C. micrantha*, presented a noticeable amount in *C. latipes* leaf oil (Figure 2).

The only description found in the literature was drastically different, with neral as a major component (24.6%), followed by an unusually high amount of undecanal (19.6%), β-phellandrene (11.4%), limonene (10.5%), and linalool (7.6%) [25].

#### 2.2.3. Related Species

The main differences between the three related species and the two sections *Papeda* and *Papedocitrus* leaf essential oils were in the proportions of γ-terpinene (6.2–28.2% vs. tr–0.2%, respectively) and *p*-cymene (4.3–11.4% vs. 0–3.5%), respectively. Thus, the three relatives were discriminated (Figure 2). However, each has its own chemical characteristics.

##### Ichang Lemon (*C. wilsonii* Tanaka)

The leaf oil composition of *C. wilsonii* was characterized by the association of γ-terpinene (19.5%), geranial (15.6%), neral (11.6%), and β-pinene (9.7%). Previously reported chemical compositions were drastically different: (i) γ-terpinene (12.9%), thymol (9.8%), β-pinene (8%), (*E*)-β-ocimene (6.9%), and *p*-cymene (4.5%) [19]; (ii) linalool (38.2%), γ-terpinene (25.4%), *p*-cymene (14.6%), neryl acetate (12.5%), β-pinene (9.8%), and nerol (5.8%) [19]; and iii) citronellol (16.9%), followed by neryl acetate (10.4%), γ-terpinene (9.9%), citronellal (9.4%), and β-pinene (6.7%) [44]. Lota et al. [37] described a similar chemical profile with quantitative variations: γ-terpinene (36.1%), geranial (3.4%), neral (2.3%), and β-pinene (14%). Taken together, these varying profiles suggest genetic variation between these representatives of Ichang lemon.

##### Yuzu (*C. junos* Sieb. Ex Tan)

Fresh leaves of *C. junos* produced an essential oil composed of γ-terpinene (28.2%), *p*-cymene (11.4%), β-phellandrene (11.2%), and linalool (10.4%). This oil also showed appreciable amounts of *p*-cymenene (6.2%), (*E*)-β-ocimene (5.0%), α-pinene (4.8%), limonene (4.7%), and β-pinene (4.1%). It could be pointed out that this oil also exhibited 2,5-dimethoxy-*p*-cymene (1.4%), a compound not identified in other *Papeda* accessions.

The chemical composition of *C. junos* leaf oil is known to be highly variable [23], and was also variable in a survey of chemical composition of 110 *Citrus* species [19]. Ten cultivars of *C. junos* were investigated, showing very different profiles dominated by (i) methyl-N-methyl anthranilate (a compound found in high quantities in *Citrus reticulata* mandarins), or (ii) γ-terpinene, in proportions varying from 22.6 to 53.2%. Three accessions in this study exhibited a composition very similar to ours, with γ-terpinene (25.7–26.6%), *p*-cymene (11.5–12.8%), β-phellandrene (8.2–12.0%), and linalool (5.8–8.1%). Another study identified the same major components but in a different ranking with 25.4% linalool, 15.6% γ-terpinene, 11.2% β-phellandrene, and 9.5% *p*-cymene [45].

This chemical composition dominated by the association γ-terpinene/*p*-cymene/linalool is frequently reported for mandarin leaf essential oils such as Wase and Owari satsumas (*C. unshiu*), Fuzhu (*C. eryhtrosa*), Kunembo (*C. nobilis*), Szibat (*C. suhuiensis*), and Sunki (*C. sunki*) [46]. This characteristic of the chemical profile of Yuzu may be inherited from its paternal lineage, the mandarin.

##### Alemow (*C. macrophylla* Wester)

The leaf oil of *C. macrophylla* was characterized by large amounts of geranial (24.7%), neral (18.9%) and limonene (17.7%) with smaller proportions of γ-terpinene (6.2%), *p*-cymene (4.3%), linalool (4.3%), and citronellal (3.5%).

The same major components were identified in another study, but in different relative quantities: limonene (31.4%), geranial (22.8%), neral (16.1%), and citronellal (13.9%), followed by δ-3-carene (3.5%) and α-terpinene (3.4%) [25].

This type of composition dominated by the association geranial/neral/limonene is usually found in leaf essential oils of citrons and some limes [37]. This characteristic of the chemical profile of Alemow might be inherited from its citron male paternal lineage.

The high diversity within the chemical composition of *Citrus* subgenus *Papeda* leaf essential oils is illustrated by a three-dimensional PCA (Figure 2) where more than 80% of the global variability is represented by the three axes. We have also visualized the chemical compounds implicated in this representation (Appendix A).

## 3. Materials and Methods

### 3.1. Plant Material

According to the systematics of Swingle and Reece (1967), ten accessions were selected to represent the diversity of the subgenus *Papeda*, including: three accessions of Ichang papeda (*C. ichangensis* Swing.) and one of Khasi papeda (*C. latipes* (Swing.) Tan.) for section *Papedocitrus*, Biasong (*C. micrantha* Wester), Combava (*C. hystrix* D.C.), and Melanesian papeda (*C. macroptera* Montr.) for section *Papeda*. Three other *Citrus* species related to *Papeda* have been added: Ichang lemon (*C. wilsonii* Tan.; *C. maxima* × *C. junos*), Alemow (*C. macrophylla* Wester; *C. micrantha* × *C. medica*), and Yuzu (*C. junos* Sieb. ex Tan.; *C. ichangensis* × *C. reticulata*) (Table 3). All the trees are maintained in the INRAE-CIRAD citrus collection (certified as Biological Resource Center (BRC) citrus NF96-600) located in San Ghjulianu, Corsica (France): latitude 42°17’ N; longitude 9°32’ E; Mediterranean climate; average: rainfall and temperature 840 mm and 15.2 °C per annum, respectively; soil derived from alluvial deposits and classified as fersiallitic; pH range 6.0–6.6 [31].

About 100 g of fruit peels and 200 g of leaves were randomly collected all around the tree. The fresh materials underwent hydrodistillation for three hours using a Clevenger type apparatus. Since peel oil yields were influenced by the presence of variable amounts of albedo during the peeling of the epicarp, they were not calculated. Distillation yields of leaf oils were calculated using the weight of essential oil/weight of fresh leaves ratio. Each sample was analyzed by gas chromatography (GC) and gas chromatography coupled with mass spectrometry (GC-MS) in order to determine the chemical composition. To avoid any misidentification, some samples, selected on the basis of the chromatogram profile, were analyzed with carbon-13 nuclear magnetic resonance (^13^C NMR) following a methodology developed in our laboratory [47].

### 3.2. Gas Chromatography (GC) Analysis

GC analyses were performed on a Clarus 500 FID gas chromatograph (PerkinElmer, Courtaboeuf, France) equipped with two fused silica gel capillary columns (50 m × 0.22 mm, film thickness 0.25 μm), BP-1 (polydimethylsiloxane) and BP-20 (polyethylene glycol). The oven temperature was programmed from 60 to 220 °C at 2 °C/min and then held isothermal at 220 °C for 20 min, injector temperature: 250 °C; detector temperature: 250 °C; carrier gas: hydrogen (1.0 mL/min); split: 1/60. The relative proportions of the oil constituents were expressed as percentages obtained by peak area normalization, without using correcting factors. Retention indices (RIs) were determined relative to the retention times of a series of n-alkanes (C_7_–C_28_) with linear interpolation (“Target Compounds” software of PerkinElmer). The EOs samples (50 mg) were diluted in chloroform (1 mL).

### 3.3. Mass Spectrometry

The EOs were analyzed with a PerkinElmer TurboMass detector (quadrupole, Perkin Elmer, Courtaboeuf, France), directly coupled to a PerkinElmer Autosystem XL, equipped with a fused silica gel capillary column (50 m × 0.22 mm *i.d*.; film thickness 0.25 µm), BP-1 (polydimethylsiloxane). Carrier gas: helium at 0.8 mL/min; split: 1/75; injection volume: 0.5 µL; injector temperature: 250 °C; oven temperature programmed from 60 to 220 °C at 2 °C/min and then held isothermal (20 min); ion source temperature: 250 °C; energy ionization: 70 eV; electron ionization mass spectra were acquired over the mass range 40–400 Da. Oil samples were diluted in deuterated chloroform with 50 mg of essential oil in chloroform (1 mL).

### 3.4. NMR Analysis

^13^C NMR analyses were performed on an AVANCE 400 Fourier Transform spectrometer (Bruker, Wissembourg, France) operating at 100.623 MHz for ^13^C, equipped with a 5 mm probe, in CDCl_3_, with all shifts referred to internal tetramethylsilane (TMS). ^13^C NMR spectra were recorded with the following parameters: pulse width (PW): 4 µs (flip angle 45°); acquisition time: 2.73 s for 128 K data table with a spectral width (SW) of 220.000 Hz (220 ppm); CPD mode decoupling; digital resolution 0.183 Hz/pt. The number of accumulated scans ranged from 2000–3000 for each sample (around 40 mg of oil in 0.5 mL of CDCl_3_). Exponential line broadening multiplication (1.0 Hz) of the free induction decay was applied before Fourier Transformation.

### 3.5. Identification of Individual Components

Identification of the components was based on: (i) comparison of their GC retention indices (RIs) on polar and apolar columns, determined relative to the retention times of a series of *n*-alkanes with linear interpolation (“Target Compounds” software of Perkin Elmer, Courtaboeuf, France), with those of authentic compounds [48]; (ii) computer matching against commercial mass spectral libraries [49,50] and by comparison of spectra with literature data [51,52]; and (iii) comparison of the signals in the ^13^C NMR spectra of EOs with those of reference spectra compiled in the laboratory spectral library, using custom-made software [47,53,54]. In the investigated samples, individual components were identified by NMR at contents as low as 0.5%.

### 3.6. Statistical Analysis

The data of investigated samples of peel and leaf essential oils were submitted to Principal Component Analysis (PCA) using PAST (Paleontological Statistics Software Package) 3.14 version software [55]. Only constituents in a proportion higher than 2% at least in one sample were used as variables for the PCA analysis.

## 4. Conclusions

We analyzed the chemical composition of peel and leaf essential oils of seven *Citrus* species belonging to two sections of the *Papeda* group. Among them, the major components of leaf essential oil were: citronellal for *C. hystrix* and *C. micrantha*; sabinene, linalool and β-pinene for *C. macroptera*; (*E*) and (*Z*)-ocimene for two accessions of *C. ichangensis*, and sabinene for the third accession; limonene, linalool, and citronellal for *C. latipes*. In the three related species, leaf oil profiles were dominated by γ-terpinene, geranial, neral, and β-pinene for *C. wilsonii*; γ-terpinene, β-phellandrene, and *p*-cymene for *C. junos*; and finally geranial, neral and limonene for *C. macrophylla*. Limonene was the major component in almost all peel oil samples, except in *C. micrantha* and *C. hystrix* oils, where β-pinene dominated. The two sections of *Papeda* are clearly distinguishable in both their leaf and fruit peel essential oil composition, supporting the classification of Swingle, who proposed two sections in *Papeda*. Furthermore, as stated in this classification, the section *Papedocitrus* is an intermediate between the two subgenera *Papeda* and *Citrus*, and certain aromatic compounds, such as limonene content, seem to confirm this status.

Substantial chemical diversity was also observed in leaf oils and peel oils between representatives of each section. However, some species (Biasong and Combava) have very similar chemical profiles, reflecting their close genetic relationship. On the other hand, the three representatives of *C. ichangensis* present very different profiles. These results suggest that *Papeda* may be an important source of aroma diversity, which may be uncovered by further surveys. The three *Citrus* × *Papeda* hybrids demonstrate that crosses between these two taxa can create high variability in the aromatic composition of essential oils. Future research may also be able to exploit this aromatic diversity by crossing these little-known citrus fruits with field crop species.

## Figures and Tables

**Figure 1 plants-10-01117-f001:**
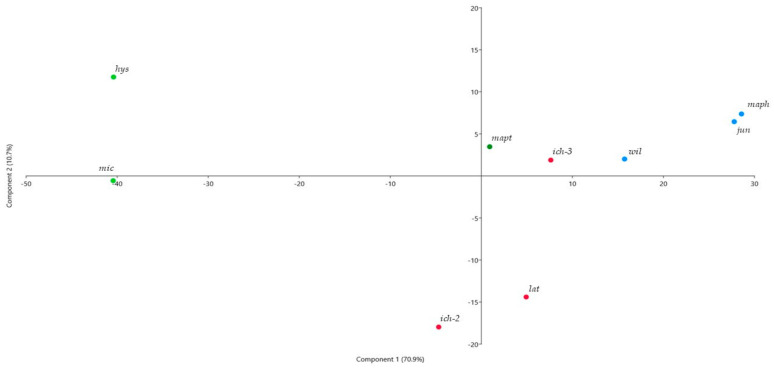
Principal Component Analysis performed on peel oil samples (components higher than 2%). Green: *hys*: *C. hystrix*, *mic*: *C. micrantha*, *mapt*: *C. macroptera*; Red: *lat*: *C. latipes*, *ich*: *C. ichangensis*; Blue: *wil*: *C. wilsonii*, *jun*: *C. junos*, *maph*: *C. macrophylla*.

**Figure 2 plants-10-01117-f002:**
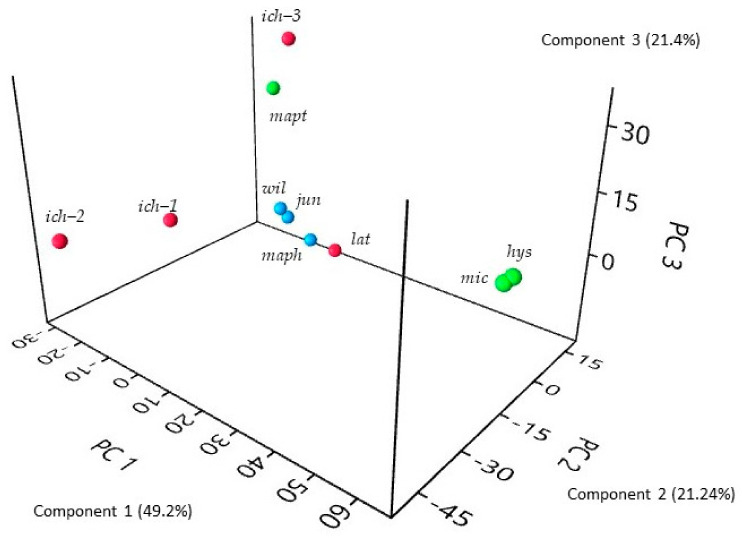
Three-dimensional Principal Component Analysis of leaf oil samples (components higher than 2%). Green: *hys*: *C. hystrix*, *mic*: *C. micrantha*, *mapt*: *C. macroptera*; Red: *lat*: *C. latipes*, *ich*: *C. ichangensis*; Blue: *wil*: *C. wilsonii*, *jun*: *C. junos*, *maph*: *C. macrophylla*.

**Table 1 plants-10-01117-t001:** Chemical composition of peel essential oils of nine *Papeda* oil samples.

N°	RI _A_	RI _P_	Name	*hys*	*mic*	*mapt*	*ich-2*	*ich-3*	*lat*	*wil*	*jun*	*maph*
1	923	1022	α-thujene	0.2	0.1	0.1	-	0.1	1.0	0.4	0.4	0.3
2	931	1020	α-pinene	3.0	2.4	1.3	-	2.1	2.4	2.2	1.6	1.0
3	945	1070	camphene	0.2	0.3	-	-	-	tr	tr	-	-
4	966	1127	sabinene	22.7	1.0	12.4	0.1	9.6	0.3	0.1	0.1	0.1
5	972	1116	β-pinene	35.0	33.4	3.9	-	0.7	1.9	3.5	0.8	0.6
6	977	1221	butyl butyrate	-	-	-	2.4	-	-	-	-	-
7	981	1166	myrcene	0.9	1.0	2.1	0.2	2.1	18.8	1.6	1.9	1.6
8	996	UD	hexyl acetate	-	-	-	-	-	-	0.3	-	-
9	998	1170	α-phellandrene	-	0.1	-	-	0.3	-	0.2	0.4	-
10	1010	1185	α-terpinene	-	0.4	0.3	-	0.1	0.2	0.1	0.2	tr
11	1013	1276	*p*-cymene	0.7	0.8	1.1	0.3	4.4	6.5	7.0	1.3	2.6
12	1022	1215	β-phellandrene *	0.3	1.0	0.5	-	8.8	0.2	3.9	2.6	0.1
13	1022	1205	limonene *	25.2	20.7	53.8	42.3	58.2	50.4	66.9	79.9	81.4
14	1026	1237	(*Z*)-β-ocimene	-	-	-	-	tr	-	-	-	0.5
15	1037	1255	(*E*)-β-ocimene	-	0.4	0.2	-	0.1	0.1	0.2	0.4	0.8
16	1049	1251	γ-terpinene	-	1.3	1.0	-	1.0	16.2	10.1	8.8	5.0
17	1062	1446	*cis*-linalool oxide THF form	0.5	0.3	1.2	-	tr	-	tr	-	0.1
18	1075	1474	*trans*-linalool oxide THF form	0.3	0.2	0.6	-	-	-	-	-	0.1
19	1079	1288	terpinolene	-	1.1	0.3	-	0.3	0.7	0.5	0.4	0.2
20	1086	1551	linalool	0.9	2.2	11.8	0.4	0.7	0.1	0.3	1.0	0.3
21	1111	1565	*cis*-*p*-menth-2-en-1-ol	tr	-	0.2	-	0.3	-	-	-	-
22	1133	1574	isopulegol	-	1.5	-	-	-	-	-	-	-
23	1133	1483	citronellal	3.4	1.5	-	-	-	-	-	-	tr
24	1144	1567	isoneopulegol	-	1.0	-	-	-	-	-	-	-
25	1160	1672	cryptone	-	-	-	-	0.4	-	0.1	-	-
26	1163	1604	terpinen-4-ol	1.2	3.8	4.3	tr	7.3	0.1	0.5	0.1	0.1
27	1175	1699	α-terpineol	0.8	6.6	0.4	0.3	0.5	0.2	0.7	0.1	0.2
28	1173	1415	butyl hexanoate	-	-	-	1.5	-	-	-	-	-
29	1175	1417	hexyl butyrate	-	-	-	0.8	-	-	-	-	-
30	1199	1837	*trans*-carveol	0.2	-	-	0.6	-	-	tr	-	tr
31	1212	1769	citronellol	0.1	6.8	-	-	-	-	-	-	-
32	1217	1683	neral	-	-	-	-	-	-	-	-	0.5
33	1237	1851	geraniol	tr	1.0	0.1	-	-	-	-	-	-
34	1245	1753	geranial	-	-	-	-	-	-	-	-	0.6
35	1309	2275	limonene-1,2-diol	-	-	-	0.6	tr	-	0.1	-	-
36	1333	1697	α-terpinyl acetate	-	-	-	-	-	-	0.3	-	-
37	1334	1664	citronellyl acetate	-	3.1	-	-	-	-	-	-	-
38	1361	1759	geranyl acetate	1.1	2.1	-	-	0.2	tr	tr	-	0.1
39	1369	1611	hexyl hexanoate	0.1	-	-	0.5	-	-	-	-	-
40	1370	1614	butyl octanoate	-	-	-	0.8	-	-	-	-	-
41	1375	1492	α-copaene	0.3	0.2	0.4	0.1	-	-	-	-	tr
42	1387	1591	β-elemene	0.2	0.2	0.4	-	tr	-	-	-	0.1
43	1410	1569	*cis*-α-bergamotene	-	-	-	0.3	-	-	-	-	tr
44	1417	1597	(*E*)-β-caryophyllene	0.1	0.1	1.7	-	-	0.1	-	-	0.3
45	1417	1572	α-santalene	-	-	-	-	0.5	-	-	-	-
46	1432	1586	*trans*-α-bergamotene	-	-	-	3.2	-	-	-	-	0.6
47	1447	1667	(*E*)-β-farnesene	-	-	-	1.8	-	-	0.2	0.1	tr
48	1469	1688	γ-muurolene	-	-	-	3.1	-	-	-	-	-
49	1475	1708	germacrene D	-	0.2	0.3	-	0.1	0.7	-	-	0.6
50	1481	1718	β-selinene	-	tr	-	2.0	-	-	-	-	-
51	1488	1718	valencene	-	-	-	2.7	-	-	0.1	-	-
52	1490	1723	α-selinene	tr	tr	0.1	0.7	-	-	-	-	-
53	1495	1750	(*E*,*E*)-α-farnesene	-	1.7	-	-	-	-	-	-	0.1
54	1500	1727	β-bisabolene	-	-	-	18.4	-	-	-	-	0.9
55	1505	1758	γ-cadinene	-	-	-	1.1	-	-	-	-	-
56	1513	1757	δ-cadinene	0.1	0.3	0.5	-	-	-	tr	-	0.1
57	1548	2043	(*E*)-nerolidol	-	tr	0.2	1.6	0.9	-	-	-	-
58	1550	1826	germacrene B	-	0.3	-	-	-	-	-	-	tr
59	1611	2254	alismol	-	0.3	0.1	1.0	-	-	-	-	0.1
60	1641	2229	intermedeol	-	-	-	4.7	-	-	-	-	-
			Monoterpene hydrocarbon	88.2	63.7	76.9	42.8	87.7	98.7	96.7	98.7	94.3
			Oxygenated monoterpene	8.5	30.1	18.6	1.9	9.4	0.4	2.0	1.2	2.0
			Sesquiterpene hydrocarbon	0.7	3.2	3.4	33.4	0.6	0.7	0.3	0.1	2.6
			Oxygenated sesquiterpene	0.0	0.3	0.3	7.3	0.9	0.0	0.0	0.0	0.1
			Acyclic compound	0.1	0.0	0.0	6.0	0.0	0.0	0.3	0.0	0.0
			TOTAL	97.5	97.3	99.3	91.5	98.6	99.9	99.3	99.9	98.9

Order of elution and relative percentages of individual components are given on an apolar column (BP-1) except those with an asterisk (*), for which percentages were taken on polar column (BP-20); RI_A_. RI_P_: retention indices measured on apolar and polar capillary columns, respectively; “-”: absence of the component; tr: trace level (<0.05%); *hys*: *C. hystrix*, *mic*: *C. micrantha*, *mapt*: *C. macroptera*, *lat*: *C. latipes*, *ich*: *C. ichangensis*, *wil*: *C. wilsonii*, *jun*: *C. junos*, *maph*: *C. macrophylla*.

**Table 2 plants-10-01117-t002:** Chemical composition of leaf essential oil of ten *Papeda* oil samples.

N°	RI _A_	RI _P_	Name	*hys*	*mic*	*mapt*	*ich-1*	*ich-2*	*ich-3*	*lat*	*wil*	*jun*	*maph*
1	923	1022	α-thujene	tr	-	0.3	-	tr	0.5	-	0.8	2.0	0.2
2	931	1020	α-pinene	0.1	-	2.0	tr	tr	2.2	0.1	2.0	4.8	0.5
3	965	1342	6-methyl-5-hepten-2-one	-	-	-	-	-	-	-	0.3	-	0.5
4	966	1127	sabinene	3.0	tr	32.4	-	tr	44.6	0.2	1.4	0.7	0.1
5	972	1116	β-pinene	0.5	0.1	15.7	-	tr	1.6	0.3	9.7	4.1	0.5
6	981	1166	myrcene	0.5	0.4	1.8	1.7	1.1	2.3	0.7	0.8	1.4	0.6
7	998	1170	α-phellandrene	-	-	0.2	0.1	0.1	1.3	-	0.3	1.7	-
8	1006	1153	δ-3-carene	tr	-	1.9	0.4	0.7	-	-	-	-	tr
9	1010	1185	α-terpinene	0.1	-	1.0	-	-	2.2	-	0.3	0.8	tr
10	1013	1276	*p*-cymene	tr	-	0.1	-	tr	0.2	0.2	5.1	11.4	4.3
11	1022	1215	β-phellandrene *	0.1	-	0.7	0.1	0.1	11.7	tr	3.4	11.2	-
12	1022	1205	limonene *	2.4	0.1	2.7	0.4	0.1	3.1	41.0	4.0	4.7	17.7
13	1026	1237	(*Z*)-β-ocimene	tr	0.1	1.8	13.0	18.2	0.7	0.1	0.1	0.2	0.3
14	1037	1255	(*E*)-β-ocimene	0.2	0.6	8.6	32.4	62.7	3.7	2.2	3.0	5.1	0.6
15	1049	1251	γ-terpinene	0.5	-	1.6	tr	-	3.5	0.1	19.5	28.2	6.2
16	1057	1467	*trans*-sabinene hydrate	tr	tr	0.2	-	-	0.6	-	-	tr	tr
17	1062	1446	*cis*-linalool oxide THF form	tr	tr	1.1	-	tr	0.2	0.1	tr	-	0.2
18	1073	1442	*p*-cymenene	-	-	-	-	tr	-	tr	-	6.2	-
19	1075	1474	*trans*-linalool oxide THF form	tr	-	0.6	-	tr	0.1	tr	-	-	0.1
20	1079	1288	terpinolene	0.1	tr	0.7	0.3	tr	0.8	tr	0.9	2.0	0.2
21	1086	1551	linalool	3.4	1.2	18.2	9.3	0.2	1.0	24.6	6.1	10.4	4.3
22	1087	1550	*cis*-sabinene hydrate	-	-	-	-	-	0.5	-	-	-	-
23	1111	1565	*cis*-*p*-menth-2-en-1-ol	-	-	0.2	-	-	0.5	-	-	-	-
24	1117	1375	allo-ocimene	-	-	-	0.4	0.4	-	-	-	-	-
25	1126	1630	*trans*-*p*-menth-2-en-1-ol	-	-	0.1	-	-	0.3	-	-	0.6	-
26	1133	1574	isopulegol	0.8	0.9	-	-	-	-	0.2	-	-	0.1
27	1133	1483	citronellal	78.1	76.1	-	-	-	0.3	14.1	1.0	-	3.5
28	1145	1567	isoneopulegol	0.3	0.3	-	-	-	-	tr	-	-	-
29	1159	UD	isogeranial	-	-	-	-	-	-	-	0.2	tr	0.4
30	1163	1604	terpinen-4-ol	0.3	-	3.8	tr	tr	8.4	-	0.4	0.2	0.2
31	1175	1699	α-terpineol	0.1	-	0.2	3.1	tr	0.3	0.1	0.3	0.1	0.6
32	1212	1769	citronellol	3.4	4.4	-	-	-	0.1	1.8	0.2	-	0.1
33	1212	1804	nerol	0.1	-	-	0.8	-	0.1	0.1	2.3	-	0.2
34	1215	1597	thymyl methyl oxide	-	-	-	-	-	-	-	-	0.3	-
35	1217	1683	neral	tr	-	-	-	-	0.1	0.8	11.6	-	18.9
36	1237	1851	geraniol	0.6	1.2	-	2.3	tr	0.3	0.1	0.4	-	0.6
37	1241	1560	linalyl acetate	-	-	-	10.8	-	-	0.1	-	-	0.2
38	1245	1753	geranial	0.1	-	-	-	tr	0.2	1.0	15.2	-	24.7
39	1268	2192	thymol	-	-	-	-	-	-	-	-	1.1	-
40	1303	1697	methyl geranate	-	-	-	-	-	0.7	-	-	-	-
41	1334	1664	citronellyl acetate	0.7	5.1	-	-	-	0.3	1.0	-	-	0.4
42	1335	1472	δ-elemene	-	0.3	-	-	-	-	-	0.3	0.1	0.1
43	1343	1728	neryl acetate	0.1	tr	-	1.2	-	0.2	0.1	1.1	-	0.5
44	1361	1759	geranyl acetate	1.2	2.9	-	2.4	-	5.0	0.1	0.1	-	2.1
45	1375	1492	α-copaene	0.2	0.3	-	0.1	tr	-	tr	-	-	0.1
46	1387	1591	β-elemene	tr	0.5	-	-	tr	-	tr	0.3	tr	0.3
47	1399	1872	2,5-dimethoxy-*p*-cymene	-	-	-	-	-	-	-	-	1.4	-
48	1417	1597	(*E*)-β-caryophyllene	1.1	0.8	0.5	1.4	-	-	2.9	0.1	0.1	2.7
49	1427	1638	γ-elemene	tr	-	-	-	0.3	-	-	-	tr	-
50	1432	1586	*trans*-α-bergamotene	0.1	-	-	0.1	0.1	-	1.1	-	-	0.6
51	1447	1667	(*E*)-β-farnesene	-	0.1	-	0.1	0.3	-	-	tr	tr	tr
52	1449	1667	α-humulene	0.1	0.1	0.1	0.2	0.2	-	0.2	0.1	tr	0.3
53	1469	1688	γ-muurolene	-	0.1	-	0.2	0.3	-	-	-	-	tr
54	1471	1668	guaia-6,10(14)-diene	-	-	-	0.3	0.4	-	-	-	-	-
55	1475	1708	germacrene D	0.1	0.2	-	-	-	0.1	0.6	1.2	tr	0.9
56	1481	1718	β-selinene	-	-	-	0.4	0.5	-	tr	-	-	tr
57	1490	1723	α-selinene	-	-	-	tr	0.3	-	-	-	-	-
58	1490	1732	bicyclogermacrene	0.3	0.2	0.4	-	-	0.1	0.3	0.1	0.1	0.4
59	1495	1750	(*E*,*E*)-α-farnesene	0.2	0.8	0.3	-	0.4	0.1	-	-	-	0.2
60	1500	1727	β-bisabolene	0.1	-	-	0.7	1.2	-	1.5	0.1	-	0.9
61	1513	1757	δ-cadinene	0.3	0.3	0.1	0.5	0.3	-	tr	-	tr	0.2
62	1534	2079	β-elemol	tr	0.1	-	-	-	-	-	1.1	-	-
63	1548	2043	(*E*)-nerolidol	0.2	tr	0.3	0.5	0.6	1.3	tr	0.3	tr	tr
64	1549	1825	germacrene B	-	0.5	-	0.4	0.6	tr	-	0.4	tr	0.2
65	1563	2121	spathulenol	-	-	0.1	0.1	tr	0.1	0.1	0.1	-	0.7
66	1570	1978	caryophyllene oxide	tr	-	-	1.8	-	-	0.2	-	-	0.4
67	1592	2033	humulene oxide II	-	-	-	0.3	0.2	-	-	-	-	tr
68	1611	2254	alismol	-	0.1	-	1.7	1.6	0.1	-	0.4	-	0.3
69	1616	2197	eremoligenol	-	-	-	0.4	-	-	0.1	0.3	-	-
70	1618	2176	γ-eudesmol	-	tr	-	0.1	-	-	tr	0.6	-	-
71	1625	2169	τ-cadinol	-	-	0.1	0.3	0.1	-	-	0.2	tr	0.1
72	1634	2225	β-eudesmol	-	-	-	0.7	0.8	-	0.2	0.6	tr	-
73	1639	2216	α-eudesmol	-	-	-	1.1	0.1	-	tr	0.5	-	-
74	1651	2145	β-bisabolol	-	-	-	0.4	-	-	-	-	-	-
75	1668	2215	α-bisabolol	-	-	-	0.4	0.5	-	0.1	-	-	0.1
76	2098	2610	(*E*)-phytol	-	0.3	0.7	3.0	1.4	tr	1.1	0.2	0.1	0.5
			Monoterpene hydrocarbon	7.4	1.3	71.5	48.6	83.4	78.3	44.9	51.2	84.4	31.1
			Oxygenated monoterpene	88.9	92.2	24.4	30.0	0.2	19.1	44.1	39.1	14.1	57.2
			Sesquiterpene hydrocarbon	2.5	4.2	1.3	4.3	4.8	0.3	6.6	2.4	0.3	7.0
			Oxygenated sesquiterpene	0.2	0.2	0.6	7.7	3.9	1.5	0.6	4.1	0.0	1.5
			Oxygenated diterpene	0.0	0.3	0.7	3.0	1.4	0.0	1.1	0.2	0.1	0.5
			TOTAL	99.0	98.3	98.5	93.6	93.8	99.3	97.2	97.1	98.9	97.3
			Yields (%; *w*/*w*)	0.05	0.02	0.02	0.18	0.17	0.08	0.08	0.03	0.04	0.10

Order of elution and relative percentages of individual components are given on an apolar column (BP-1) except those with an asterisk (*), for which percentages were taken on polar column (BP-20); RI_A_. RI_P_: retention indices measured on apolar and polar capillary columns, respectively; “-”: absence of the component; tr: trace level (<0.05%); *hys*: *C. hystrix*, *mic*: *C. micrantha*, *mapt*: *C. macroptera*, *ich*: *C. ichangensis*, *lat*: *C. latipes*, *wil*: *C. wilsonii*, *jun*: *C. junos*, *maph*: *C. macrophylla*.

**Table 3 plants-10-01117-t003:** List of studied species and accessions.

Scientific Name	Common Name	Sample	ICVN
*C. hystrix* DC.	Combava	*hys*	0100630
*C. macroptera* Montr.	Melanesian papeda	*mapt*	0100686
*C. micrantha* Wester	Biasong	*mic*	0101140
*C. ichangensis* Swingle	Ichang papeda	*ich-1*	0100687
*C. ichangensis* Swingle	Ichang papeda	*ich-2*	0110241
*C. ichangensis* Swingle	Ichang papeda	*ich-3*	0110240
*C. latipes* (Swingle) Tanaka	Khasi papeda	*lat*	0110243
*C. wilsonii* Tanaka	Ichang lemon	*wil*	0100844
*C. junos* Siebold ex Tanaka	Yuzu	*jun*	0100988
*C. macrophylla* Wester	Alemow	*maph*	0110058

ICVN: International Citrus Varietal Numbering.

## Data Availability

Data is contained within the article or Appendix A.

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
