# Peer review of "Chemical Variability of Peel and Leaf Essential Oils in the Citrus Subgenus Papeda (Swingle) and Few Relatives"

_plants, 2021, doi:10.3390/plants10061117_

Round 1

Reviewer 1 Report

You have used many resources in this well-founded article. Was C-NMR analysis also required?

Reviewer 2 Report

  • the section "materials and methods" should follow immediately after the introduction and purpose of the work, and not at the end of the article
  • you should carefully draw up a list of references according to the rules of the journal
  • In the section "Introduction" - to give only information about the published papers. Achievements and results of the authors, in a brief form can be given in the abstract, and fully given - in the "conclusion".

Reviewer 3 Report

Describe how the data were prepared for the PCA.  Were they median centered or any analyzed as raw data?  How were any absent data addressed?  Is this particular form of PCA robust or sensitive to outliers in the data set? Given the high variance in the phenotypic expression of terpene synthesis and the high heterogeneity in the genetic makeup please address why single bulk-composite samples are adequate to address differences across the cultivars using PCA.  Seems difficult to make a definitive cluster analysis with n=1. 

For figure 2 rotate the plot to provide a better view from the top of the third dimension. 
